# *DOT1L-HES6* fusion drives androgen independent growth in prostate cancer

Matti Annala[1,2,*], Kati Kivinummi[1,*], Katri Leinonen[1], Joonas Tuominen[1], Wei Zhang[3], Tapio Visakorpi[1,†] & Matti Nykter[1,†]

Molecular therapies targeting the androgen receptor (*AR*) or pathways involved in androgen synthesis form a critical component of the standard-of-care in treating aggressive, non-localized prostate cancers. The major problem with these therapies is that castration resistant clones arise within 1–3 years of treatment initiation, leading to clinical relapse and eventual death. Previously reported mechanisms of castration resistance include amplification and mutation of *AR* (Taplin *et al*, 1995; Visakorpi *et al*, 1995), neuroendocrine differentiation (Beltran *et al*, 2011), and aberrant activation of the glucocorticoid receptor (Arora *et al*, 2013). In a previous issue of this journal, Ramos-Montoya *et al*, 2014 implicated the transcription factor *HES6* as another important player in the induction of castration resistance. In this correspondence, we present further evidence for the role of *HES6* in castration resistant prostate cancer. We report a case of *AR*-negative prostate cancer driven by a *DOT1L-HES6* fusion gene which directly induces overexpression and pathological activation of *HES6*.

We set out to study late stage prostate cancer by performing whole transcriptome and genome sequencing of two *AR*-negative prostate cancers from distinct patients. Sample #1 was obtained at prostatectomy from a 53-year-old patient with a Gleason 5 + 5 non-metastatic tumor and a prostate specific antigen (PSA) serum level of 4.8 μg/l. The tumor cells expressed high levels of *ASCL1, CHGA, SYP,* and *HES6,* four classical markers of neuroendocrine prostate cancer

(Fig 1A) (Beltran *et al*, 2011). Sample #2 was obtained by transurethral resection of the prostate (TURP) from a 70-year-old patient originally diagnosed with a Gleason 4 + 5 non-metastatic tumor with a PSA of 62 μg/l. The diagnostic biopsy was positive for AR and ERG expression and negative for CHGA (Fig 1B). The patient was treated with orchiectomy immediately after diagnosis and did not undergo prostatectomy. The TURP sample was taken 13 months after orchiectomy and was negative for AR and ERG expression. The patient had a positive bone scan and a PSA of 1.1 μg/l when the TURP was performed, and died of his cancer 1 month later (14 months after orchiectomy). Interestingly, the TURP sample did not show elevated expression of *CHGA, SYP,* or *ASCL1,* but did show strong *HES6* expression (Fig 1A). Both samples were negative for *MYCN* and *AURKA* amplification.

To study the TURP sample further, we used ChimeraScan (Iyer *et al*, 2011) and an in-house algorithm to search for evidence of gene fusions in the transcriptome and whole genome sequencing data. Both algorithms identified a novel *DOT1L-HES6* fusion gene, caused by an interchromosomal rearrangement that fused intron 9 of *DOT1L* with a position 4 kb upstream of *HES6*, resulting in *HES6* overexpression (Fig 1C). HES6 is a member of the basic helix-loop helix (bHLH) family of transcription factors, and its expression is driven by ASCL1 in differentiating neurons (Nelson *et al*, 2009; Webb *et al*, 2013). *HES6* was highly expressed in neuroendocrine prostate cancer models NCI-H660, LuCaP-49, and LuCaP-93, with concomitant

high *ASCL1* expression (Fig 1A). Among all AR-negative tumors we tested, the *DOT1L-HES6* positive TURP sample from patient #2 was unique in having high *HES6* but no *ASCL1* activity (Fig 1A). This led us to hypothesize that the *DOT1L-HES6* fusion results in ASCL1-independent activation of HES6, which in turn promotes androgen independent growth. To test whether *HES6* overexpression induced androgen independence, we transfected androgen responsive LNCaP cells with a *HES6* vector, resulting in 28-fold overexpression of *HES6* relative to cells transfected with empty vector ($P = 0.0173$, unpaired two-tailed *t*-test, $n = 2$) (Fig 1D). We then grew the cells in mediums with different DHT levels and observed that *HES6*-transfected cells were able to grow in DHT concentrations as low as 0 nM ($P = 9.6e-27$, two-way analysis of variance, $n = 4$) and 1 nM ($P = 4.4e-11$, two-way analysis of variance, $n = 4$), while LNCaP cells transfected with empty vector were unable to grow in DHT-depleted mediums (Fig 1D). This finding is in agreement with the *HES6* overexpression phenotype reported by Ramos-Montoya *et al.*

The diagnostic biopsy of patient #2 was negative for *DOT1L-HES6* and *HES6* expression based on qRT-PCR, indicating that the fusion gene had originated post-orchiectomy (Fig 1E). To show that the *DOT1L-HES6* positive TURP sample did not represent a new and independent tumor, we used the sequencing data to search for vestigial evidence of the *ERG* fusion present in the original diagnostic biopsy. Whole genome sequencing revealed a characteristic three

1  Institute of Biosciences and Medical Technology, Tampere, Finland
2  Department of Signal Processing, Tampere University of Technology, Tampere, Finland
3  Department of Pathology, University of Texas M.D. Anderson Cancer Center, Houston, TX, USA. E-mails: tapio.visakorpi@uta.fi, matti.nykter@uta.fi
   *Equally contributing authors
   †Co-corresponding authors
DOI 10.15252/emmm.201404210 | Published online 8 July 2014

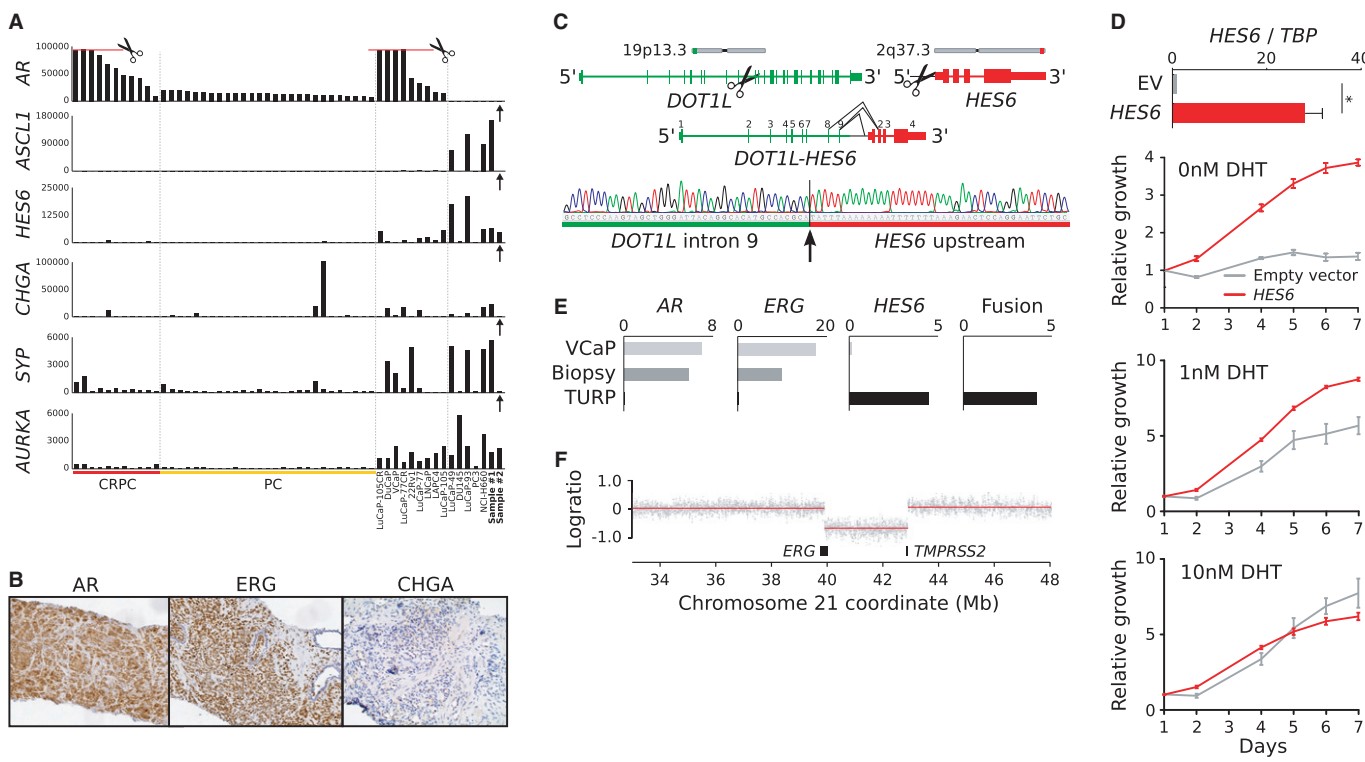

**Figure 1. *DOT1L-HES6* fusion induces androgen independent growth in prostate cancer cells.**

(A) Expression patterns of *AR*, *ASCL1*, *HES6*, *CHGA*, *SYP*, and *AURKA* in samples #1 and #2, and in an unpublished sequencing cohort of 11 CRPC TURP samples, 27 prostatectomy samples, and 14 prostate cancer cell lines and xenografts. (B) Immunostaining of AR, ERG, and CHGA in the original diagnostic needle biopsy of patient #2. (C) Structure of the *DOT1L-HES6* fusion gene identified in the TURP sample from patient #2. Black lines indicate exon-exon junctions with RNA-seq evidence. Genomic breakpoint was validated with Sanger sequencing from genomic DNA. (D) Stable transfection of *HES6* into LNCaP cells resulted in 28-fold overexpression of *HES6* ($P = 0.0173$, unpaired two-tailed *t*-test, $n = 2$). *HES6*-transfected cells maintained their growth in DHT concentrations as low as 0 nM ($P = 9.6\text{e-}27$, two-way analysis of variance, $n = 4$) and 1 nM ($P = 4.4\text{e-}11$, two-way analysis of variance, $n = 4$). Error bars, s.e.m. (E) Quantitative RT-PCR measurement of *AR*, *ERG*, *HES6*, and *DOT1L-HES6* expression in the original diagnostic needle biopsy and TURP sample of patient #2. The ERG-fusion-positive VCaP cell line is included as a control. All expression values are normalized against *TBP* expression. (F) Read coverage log ratios based on whole genome sequencing reveal the presence of a clonal *TMPRSS2-ERG* deletion in the AR-negative TURP sample from patient #2.

megabase deletion between the genes *TMPRSS2* and *ERG* in chromosome 21 in the TURP sample (Fig 1F). Transcriptome sequencing also identified residual *TMPRSS2-ERG* expression in the TURP sample, although expression was very weak due to minimal AR activity.

In their publication, Ramos-Montoya *et al* proposed a model in which HES6 promotes androgen independence by modulating AR binding. The lack of AR activity in our *DOT1L-HES6* fusion positive sample may indicate the existence of additional, AR-independent mechanisms. An alternative hypothesis is that the *DOT1L-HES6* fusion in the TURP sample from patient #2 promoted castration resistance at an intermediate stage of tumor evolution, but was later subsumed by another mechanism that additionally resulted in complete loss of AR expression. Nonetheless, the lack of *ASCL1*, *CHGA*, and

*SYP* overexpression distinguishes this tumor from classical neuroendocrine prostate cancers and highlights the role that HES6 plays in castration resistant and androgen independent tumors. This finding also calls for a more extensive search for *HES6* genomic alterations in cohorts of AR-negative and castration resistant prostate cancers.

## Acknowledgements

We wish to thank Ms. Marika Vähä-Jaakkola and Ms. Päivi Martikainen for their skillful technical assistance. We are grateful to Prof. Teuvo Tammela, University of Tampere, Finland for providing clinical samples. We are grateful to Prof. Robert L. Vessella, University of Washington, SE, USA, for providing us with LuCaP xenografts. The work was supported by grants from the Finish Funding Agency for Technology and Innovation Finland Distinguished Professor programme (MN), Academy of Finland (project no. 269474 MN, project no. 127187 TV), Sigrid Juselius Foundation (MN, TV), Emil Aaltonen Foundation (MA, MN), Competitive State Research Financing of the Expert Responsibility area of Tampere University Hospital (Grant 9N087 TV), and EU-FP7 Marie Curie Integrated Training Network, PRO-NEST (TV), the National Institutes of Health (U24CA143835, WZ).

## Author contributions

MA, KK, TV, and MN conceived and designed the experiments. MA performed computational and statistical analysis of the data. KK, KL, and JT performed wetlab experiments. MA and KK wrote the manuscript. MA, KK, WZ, TV, and MN discussed and reviewed the manuscript.

## Conflict of interest

The authors declare that they have no conflict of interest.

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
