## [Review Process File · EMBO Molecular Medicine]

DOT1L-HES6 fusion drives androgen independent growth in prostate cancer

Matti Annala, Kati Kivinummi, Katri Leinonen, Joonas Tuominen, Wei Zhang, Tapio Visakorpi, Matti Nykter

Corresponding author: Matti Nykter, Institute of Biosciences and Medical Technology

Review timeline:

Submission date:	29 April 2014
Editorial Decision:	08 May 2014
Revision received:	23 May 2014
Editorial Decision:	11 June 2014
Revision received:	13 June 2014
Accepted:	17 June 2014

Transaction Report:

Editor: Roberto Buccione

1st Editorial Decision

08 May 2014

Thank you for your correspondence manuscript submission to EMBO Molecular Medicine.

I have asked an expert on the topic to evaluate your manuscript. As you will see s/he is quite positive (please see below). However, the Reviewer does have some suggestions for additional inclusions and clarifications.

I invite you to submit a revised manuscript addressing the Reviewer's concerns with additional experimental data where appropriate.

Please note that it is EMBO Molecular Medicine policy to allow a single round of revision only and that, therefore, acceptance or rejection of the manuscript will depend on the completeness of your responses included in the next, final version of the manuscript.

We look forward to receiving your revised manuscript.

***** Reviewer's comments *****

Referee #1 (Remarks):

In general I have no major concerns with this type of additional reporting, and appreciate the attempts to support original novel papers with additional data from other groups. I am not sure of the format, but would suggest requesting some additional clarification as below

They report 2 cases of AR negative PCA- Given that this phenotype can be a neuroendocrine PC, they should mention whether they were small cell carcinomas with low PSA. A brief clinical history would be helpful.

They show AR + PCA from primary tumor of case 2 in figure but dont show the AR neg tumor they are highlighting. The primary was ERG pos - was FISH ERG pos in recurrence indicating same clone? What pipeline do they use to detect fusion from RNA-seq?

1st Revision - authors' response

23 May 2014

Dear Editor,

We would like to thank you and the reviewer for your thoughtful comments and suggestions regarding our correspondence. We have revised our manuscript to address all of the points raised by the reviewer, including new experimental data. Please find our responses to the reviewer's suggestions below.

Referee #1:

In general I have no major concerns with this type of additional reporting, and appreciate the attempts to support original novel papers with additional data from other groups. I am not sure of the format, but would suggest requesting some additional clarification as below

They report 2 cases of AR negative PCA- Given that this phenotype can be a neuroendocrine PC, they should mention whether they were small cell carcinomas with low PSA. A brief clinical history would be helpful.

We have revised the manuscript to include a brief clinical history for patient #1 (the classical neuroendocrine prostate cancer), and a more extensive clinical history for patient #2 (the *DOT1L-HES6* positive cancer), including a timeline of serum PSA levels, Gleason grades, survival times, and the times when treatments were administered. The PSA measurements for the two patients support the conclusion that the tumor of patient #1 was AR negative already at diagnosis, whereas the tumor of patient #2 was AR positive at diagnosis but became AR negative after orchiectomy.

They show AR + PCA from primary tumor of case 2 in figure but dont show the AR neg tumor they are highlighting.

The FFPE slide for the TURP sample was not available, so we could not perform immunohistochemistry for AR and ERG. However, in Figure 1a we show based on transcriptome sequencing data that AR is not expressed in the TURP sample from patient #2. We have now also performed a qRT-PCR experiment where we compared the expression of *AR*, *ERG*, *HES6* and the *DOT1L-HES6* fusion junction in the diagnostic biopsy and TURP sample from patient #2. This data is shown in Figure 1e of the revised manuscript.

The primary was ERG pos - was FISH ERG pos in recurrence indicating same clone?

In the revised manuscript, we have added Figure 1f which shows the presence of the characteristic *TMPRSS2-ERG* deletion in the TURP sample. The deletion was identified based on whole genome sequencing data. We additionally validated the presence of the *TMPRSS2-ERG* fusion using FISH, but did not include this figure in the manuscript as the whole genome sequencing data shows the deletion more conclusively.

What pipeline do they use to detect fusion from RNA-seq?

We used ChimeraScan (Iyer et al. 2011) and an in-house algorithm for fusion gene detection. Both methods successfully identified the *DOT1L-HES6* fusion gene based on transcriptome sequencing data. We have revised the manuscript to note this fact.

2nd Editorial Decision

11 June 2014

Thank you for the submission of your revised Correspondence manuscript to EMBO Molecular Medicine. We have now received the enclosed report from the Reviewer that was asked to re-assess it. As you will see the s/he is now supportive and I am pleased to inform you that we will be able to accept your manuscript pending the following final amendments:

- 1) As per our Author Guidelines, the description of all reported data that includes statistical testing must state the name of the statistical test used to generate error bars and P values, the number (n) of independent experiments underlying each data point (not replicate measures of one sample), and the actual P value for each test (not merely 'significant' or 'P < 0.05').
- 2) Please provide up to 5 keywords and a running title

I look forward to receiving your revised manuscript as soon as possible and in any case no later than two weeks from now.

***** Reviewer's comments *****

Referee #1 (Novelty/Model system Comments for Author):

They have adequately addressed the questions in the original review
No further comments or concerns from me